# Comparison of eHealth Literacy Scale (eHEALS) and Digital Health Literacy Instrument (DHLI) in Assessing Electronic Health Literacy in Chinese Older Adults: A Mixed-Methods Approach

**DOI:** 10.3390/ijerph20043293

**Published:** 2023-02-13

**Authors:** Luyao Xie, Phoenix K. H. Mo

**Affiliations:** Centre for Health Behaviours Research, JC School of Public Health and Primary Care, The Chinese University of Hong Kong, Hong Kong, China

**Keywords:** eHealth literacy, Chinese older adults, eHEALS, DHLI, mixed methods

## Abstract

This study compared the reliability, construct validity, and respondents’ preference of the Chinese version of 8-item eHEALS (C-eHEALS) and 21-item DHLI (C-DHLI) in assessing older adults’ electronic health (eHealth) literacy using a mixed-methods approach. A web-based, cross-sectional survey was conducted among 277 Chinese older adults from September to October 2021, and 15 respondents were subsequently interviewed to understand their preference of scale to use in practice. Results showed that the internal consistency and test-retest reliability of both scales were satisfactory. For the construct validity, the C-DHLI score showed stronger positive correlations with having Internet use for health information and higher educational attainments, occupational skill levels, self-rated Internet skills, and health literacy than the C-eHEALS score. In addition, younger age, higher household income, urban residence, and longer Internet use history were only positively correlated with C-DHLI score. Qualitative data suggested that most interviewees perceived the C-DHLI as more readable than C-eHEALS for its clear structure, specific description, short sentence length, and less semantic complexity. Findings revealed that both scales are reliable tools to measure eHealth literacy among Chinese older adults, and the C-DHLI seemed to be a more valid and favored instrument for the general Chinese older population based on the quantitative and qualitative results.

## 1. Introduction

The Internet is now well integrated into our daily lives. It has become the most readily accessible resource for obtaining health information, bringing many opportunities for self-care [1]. As the major users of healthcare services, older adults are one of the groups who are most likely to benefit from using the Internet for health information [2]. However, older adults are still one of the groups with the most difficulty in using digital health, which further exacerbates digital divide and health disparities [3,4]. Therefore, improving skills in searching for, understanding, and evaluating online health information, which can be reflected as eHealth literacy, would be particularly essential for them.

Electronic health (eHealth) literacy, first mentioned by Norman and Skinner in 2006, is defined as the ability of individuals to seek, find, understand, and appraise health information from electronic resources and apply such knowledge to addressing health problems [5]. The roles of eHealth literacy in improving or maintaining the health of the elderly have been more and more emphasized [6]. Different levels of eHealth literacy indicate that not everyone has the capacity to benefit from digital health [7], which may be a contributing factor to the health disparities. For example, older individuals who are low eHealth literacy may not able to take full advantage of the Internet in obtaining health information and support in their health, thereby solidifying the health disparities [8,9]. Therefore, a reliable and valid tool to measure eHealth literacy is particularly important, as it helps identify those vulnerable groups in this digital age for targeted interventions to improve their eHealth literacy.

The most widely used tool for eHealth literacy assessment is the 8-item eHealth literacy scale (eHEALS) developed by Norman and Skinner in 2006, which measures one’s perceived ability or self-efficacy to use the Internet for health information [10]. The eHEALS has been validated worldwide in at least ten languages, with good psychometric properties shown in different populations, including Korean, Norwegian, Iranian, Chinese, and US older populations [6,11]. It has thus become a well-accepted measure for eHealth literacy. In addition to eHEALS, other instruments have also been developed to assess eHealth literacy in recent years, including the digital health literacy instrument (DHLI). This scale was published in 2017 by Van der Vaart and Drossaert; it measures seven skill categories regarding using the Internet for health information, including operational skills, navigation skills, information searching, evaluating reliability, determining relevance, adding self-generated content, and protecting privacy [12]. In this scale, respondents are asked how often they experience specific situations or how difficult they find specific tasks relating to searching for health information on the Internet. This instrument was first validated among 200 Dutch adults and thereafter among 180 Korean older adults [13].

Although eHEALS is the most used instrument to measure eHealth literacy, some concerns have been raised. First, it was developed in 2006, and since then a drastic change on the Internet has been observed, including a more advanced technology and increased popularity of social media. Some scholars criticized its insufficient ability to capture the vibrant Internet characteristics (e.g., online interaction) in more recent years [14]. Indeed, in 2011, its developer Norman also mentioned that some refinements, such as adding an interactive subscale, were needed for the eHEALS [15]. In addition, the eHEALS measures individuals’ self-efficacy and comfort with health-related Internet use, but the relationship between efficacy-based eHEALS and actual performance on health-related Internet use was shown to be weak in some studies [16,17]. Thus, its ability to predict one’s actual eHealth literacy level remains questionable and needs further testing.

Compared to eHEALS, the DHLI measures both Health 1.0 and interactive Health 2.0 aspects (e.g., adding self-generated content) [12]. Moreover, all items of the DHLI strive to capture participants’ actual performance on the Internet by asking questions related to specific skills or scenarios. Therefore, the DHLI may reflect one’s eHealth literacy more objectively than self-perception using eHEALS. Previous studies have translated the eHEALS into Chinese and used it in Chinese older populations [18,19]. As the DHLI is a newly developed instrument, its psychometric performance needs to be further validated in more populations. Our research team has translated the DHLI into Chinese (C-DHLI) and evaluated its psychometric properties (e.g., reliability and factorial validity) among Chinese older adults, and the results show good psychometric properties of C-DHLI in Chinese older adults.

Due to the different emphases and characteristics of eHEALS and DHLI, more investigations are needed to compare their applicability and relevancy in assessing Chinese older adults’ eHealth literacy in practical settings. In addition to examining the reliability of the C-DHLI and C-eHEALS, the present study expanded research on eHealth literacy assessments by examining their construct validity. Typically, construct validity is established by presenting correlations between a measure of a construct and some other measures that should, theoretically, be associated with it (convergent validity) or had no association it (discriminant validity) [20]. In the current study, construct validity was assessed by testing the correlations between C-eHEALS/C-DHLI score and a set of variables in the three aspects of demographics, Internet use behaviors, and health literacy, which were closely associated with eHealth literacy among older adults in extant literature.

First, inequalities in demographics and socioeconomic status can have great influence on older adults’ eHealth usage and eHealth literacy [21]. The Integrated Model of eHealth Use (iMeHU), proposed by Bodie and Dutta in 2008, suggests macro-level disparities in social structures (e.g., demographics) are connected to health disparities through the micro-level conduits, including eHealth usage and eHealth literacy [8]. A recent systematic review also showed that socio-demographic factors, such as age, educational attainment, and socioeconomic status, were important influencing factors of eHealth literacy for Chinese older adults [22]. Second, older adults are still one of the most vulnerable groups in the digital divide [7,23]. Compared with young generations, older adults reported having lower Internet use, poorer Internet use skills, and less Internet use for health information [24]. Internet use behaviors, such as Internet use history, frequency, and health-related Internet use, were found to be predictive of eHealth literacy among older adults [21,25,26]. Third, health literacy, defined as “the degree to which individuals have the capacity to obtain, process, and understand basic health information and services needed to make appropriate health decisions [27]”, can be seen as an embedded core concept within eHealth literacy based on the Lily Model [5]. Therefore, these variables can be considered as the related constructs to measure older adults’ eHealth literacy.

In addition, given that eHealth literacy capabilities can be seen as important personal resources, it is also necessary to understand respondents’ personal preferences about the measurements. The present study will utilize qualitative interviews to determine respondents’ perceptions of the two scales. Such approach can provide important information for understanding respondents’ preferences regarding the scales used in research and clinical settings [28].

Based on the above, the purpose of this study was to compare the reliability, construct validity, and respondents’ perceptions of the C-eHEALS and C-DHLI using a mixed-methods approach. It is expected that findings of the study would help to identify a valid and reliable instrument for assessing eHealth literacy among Chinese older adults.

## 2. Materials and Methods

### 2.1. Participants and Procedures

A web-based cross-sectional survey was conducted among Chinese older adults from September to October 2021. A non-probability, convenience sample of older adults above 55 years of age in 30 provinces and municipalities of mainland China was recruited. The inclusion criteria of the participants were: (1) being aged 55 or over, (2) having Internet use experience, and (3) having cognitive ability to complete a self-reported survey. Eligible participants were invited to participate in the web-based survey via social networking sites (e.g., WeChat). The online questionnaire, which included the informed consent and study questionnaire, was accessed through a hyperlink or a QR code of *Sojump* (a major platform for online surveys in China). Participants self-administered the online questionnaire after giving their informed consent, which took around 15 min to complete. After two weeks, approximately 20% of them were randomly selected and invited to complete another questionnaire which consisted of the two eHealth literacy scales again. No incentives were given to the participants.

### 2.2. Measures

Socio-demographic characteristics included the respondents’ information about age, gender, types of residence, education, occupation (currently or before retirement), and monthly household income.

Internet use-related factors included respondents’ Internet use history (years), self-rated Internet use skills (response options from 1 = “Poor” to 5 = “Excellent”) and Internet use for health information (0 = “No”, 1 = “Yes”).

eHealth literacy was assessed by two instruments: the 8-item C-eHEALS and the 21-item C-DHLI. As introduced before, in the C-DHLI, respondents were asked how often they experience specific situations or how difficult they find it to perform specific tasks by searching for health information from the Internet. The 4-point Likert scale has response options ranging from “1 = very easy” to “4 = very difficult” or from “1 = never” to “4 = often”. Scores need to be reversed, and a higher score represents a higher level of eHealth literacy. In this study, the three optional items about protecting privacy were not included in the validation analysis due to low completion rates [12]. Another instrument—C-eHEALS—measures one’s perceived ability to use Internet for health information. Items are measured by a 5-point Likert scale with responses from “1 = strongly disagree” to “5 = strongly agree”. Total scores are summed to range from 8–40, with higher scores indicating higher perceived eHealth literacy [10].

Health literacy was assessed by a 14-item health literacy scale (HLS-14) developed by M. Suka et al., which contains three dimensions: functional, interactive, and critical [29]. The response was rated on a 5-point Likert scale (1 = “Strongly disagree” to 5 = “Strongly agree”, with higher scores indicating higher health literacy.

### 2.3. Qualitative Interviews

After the web-based survey, participants who had shown interest in taking part in the follow-up qualitative study were invited to provide their contact information. Those with various socio-demographic backgrounds were randomly selected and approached by one research team member for their consent to be interviewed. During the interviews, their perceptions and preference for the two eHealth literacy scales were asked and recorded (See Appendix A for sample interview questions). The average length of the individual interview was around 20 min.

The recruitment for interviewees stopped when data saturation emerged (i.e., where no new insights/themes/issues emerged from the interviews) [30,31]. A total of 15 respondents were finally included in the qualitative interviews.

Ethical approval was obtained from the Survey and Behavioral Research Ethics Committee of the Chinese University of Hong Kong (No. SBRE-21-0005).

### 2.4. Data Analysis

Descriptive statistics regarding respondents’ socio-demographic characteristics, Internet use behaviors, eHealth literacy scores on two scales, and health literacy were described by mean ± standard deviation (SD) for continuous variables (normality, median and quartile for non-normality) and number (percentage) for categorical variables.

#### 2.4.1. Reliability

Internal consistency of both scales was assessed by Cronbach’s coefficient alpha. A threshold of 0.7 was considered acceptable, a value >0.8 indicated good internal consistency, and a value >0.9 indicated excellent internal consistency [32]. 

Test-retest reliability was calculated by the interclass correlation coefficient (ICC), with values less than 0.5, between 0.5 and 0.75, between 0.75 and 0.9, and greater than 0.90, indicating poor, moderate, good, and excellent reliability, respectively [33].

#### 2.4.2. Construct Validity

Based on the theoretical and empirical evidence, nine a-priori hypotheses were used to evaluate the construct validity of the C-DHLI and C-eHEALS as follows. The process of construct validation test used in this study followed the methodology in examining scale construct validity used in many previous studies [34,35,36].

**H1:** *Older age would be negatively associated with eHealth literacy among Chinese older adults*.

**H2:** *Higher educational attainments would be positively associated with eHealth literacy among Chinese older adults*.

**H3:** *Urban residence would be positively associated with eHealth literacy among Chinese older adults*.

**H4:** *Higher occupational skill levels would be positively associated with eHealth literacy among Chinese older adults*.

**H5:** *Higher household income would be positively associated with eHealth literacy among Chinese older adults*.

**H6:** *Longer Internet use history would be positively associated with eHealth literacy among Chinese older adults*.

**H7:** *Higher self-rated Internet skills would be positively associated with eHealth literacy among Chinese older adults*.

**H8:** *Having used Internet for health information would be positively associated with eHealth literacy among Chinese older adults*.

**H9:** *Higher health literacy would be positively associated with eHealth literacy among Chinese older adults*.

One-way analyses of variance (ANOVA) were used to analyze responses for C-eHEALS and C-DHLI scores, with age category, types of residence, education, occupation, household income, Internet use history, self-rated Internet skills, and using the Internet for health information as group factors, followed by LSD post-hoc tests. In addition, Pearson or Spearman correlations were also used to test the above hypotheses for the effect size and direction.

#### 2.4.3. Qualitative Data Analysis

The qualitative data were transcribed and translated into English. We used inductive thematic analysis to analyze the interview data related to the respondents’ preference decisions [37]. The authors coded all transcripts and then reviewed the codes together. After the initial coding, labels were attached to text fragments that appeared to be important to the interview questions, and the analysis was carried out in an iterative fashion to develop a set of themes regarding to the interview topic (i.e., their preference between the scales). The authors compared the raw data with emerging theme label and further refined the themes by merging, adding, and removing redundant themes. The results were presented to a research team that consisted of a qualitative expert, a psychologist, and two public health postgraduate students for finalization.

## 3. Results

### 3.1. Descriptive Statistics

The characteristics of respondents are presented in Table 1. A total of 277 participants were included, and 110 (39.7%) of them were 60 years old or over. Over half (51.3%) were male, and half (50.9%) lived in cities. Most of them (58.8%) had education attainments of middle school or high school. Only 18.4% had a monthly household income of less than RMB 2500. For their Internet use, 35.7% had an Internet use history of over 10 years, and most (60.3%) rated their Internet skills as “Average”. 79.4% of participants had used the Internet for health information.

### 3.2. Reliability

Sixty-two participants completed the C-eHEALS and C-DHLI again two weeks after baseline. The internal consistency of both scales was good, with both Cronbach’s alpha of 0.94. For the test-retest reliability, both C-eHEALS and C-DHLI showed excellent test-retest reliability on their total scores by 0.92 and 0.94 of ICC between baseline and two weeks later (*p* < 0.001).

### 3.3. Construct Validity

As shown in Table 1 and Table 2, the results confirmed a-prior-defined hypotheses to evaluate the construct validity of C-eHEALS and C-DHLI. All hypotheses were fulfilled regarding the C-DHLI, whereas only five hypotheses (i.e., H2: education, H4: occupation, H7: self-rated Internet skills, H8: using the Internet for health information, and H9: health literacy) were fulfilled in the C-eHEALS. Table 2 presents the correlation coefficients of two eHealth literacy scales and key variables for the nine hypotheses, with all the directions as expected in both scales. For the C-DHLI, higher educational attainments (ρ = 0.479), higher household income (ρ = 0.403), higher self-rated Internet skills (ρ = 0.492), having used the Internet for health information (ρ = 0.432), and higher health literacy (ρ = 0.562) were moderately correlated with higher C-DHLI scores (*p* < 0.001). Specifically, the functional literacy dimension (ρ = 0.402) of health literacy showed a moderate positive correlation with C-DHLI scores but no significant correlation with C-eHEALS. Nevertheless, age (*r* = −0.217) showed a weak-to-moderate negative correlation with C-DHLI scores; unban residence (ρ = 0.240), higher occupational skill levels (ρ = 0.298), and longer Internet use history (ρ = 0.346) also showed a weak-to-moderate positive correlation with higher C-DHLI scores (*p* < 0.001).

For C-eHEALS, higher self-rated Internet skills (ρ = 0.242), having used the Internet for health information (ρ = 0.328), and higher health literacy (ρ = 0.391) were weakly-to-moderately correlated with higher C-eHEALS scores; higher educational attainments (ρ = 0.179, *p* < 0.01) and occupational skill levels (ρ = 0.150, *p* < 0.05) were weakly correlated with higher C-eHEALS scores. Other variables showed no significant correlations with C-eHEALS scores.

### 3.4. Qualitative Results

As shown in Table 3, among the 15 interviewees, five of them (33.3%) preferred to use the C-eHEALS in practical settings, labeled *A1*–*A5*, and eight (53.3%) preferred to use the C-DHLI, labeled *B6*–*B13*. Two (13.3%) did not give an explicit choice and were labeled *C14*–*C15*. Through an iterative process, the data were saturated on two themes: **item readability** and **content comprehensiveness**, that influenced respondents’ preference to use a particular scale.

#### 3.4.1. Item Readability

The scale’s readability is one of the most important factors affecting respondents’ preference for which scale to use. For the five interviewees who preferred the C-eHEALS, three interviewees (i.e., *A1*, *A2*, and *A3*) stated the items in both C-eHEALS and C-DHLI were easy to read and understand. With regards to their demographic characteristics, they generally had higher education attainment, higher household income, urban residence, longer Internet use history, having used the Internet for health information, and higher C-eHEALS and C-DHLI scores. They had no difficulty answering both scales; thus, the **scale’s brevity** (i.e., fewer item numbers) became the main reason for their choice.


*“Both scales are easy to answer, but the C-DHLI with more than twice as many items as C-eHEALS seems too long and tedious. I prefer to use the shorter one (i.e., C-eHEALS).”*

*[Interviewee A2]*


However, two other interviewees (i.e., *A4* and *A5*) found both scales quite difficult to understand and answer by themselves, especially *Interviewee A5* who could hardly answer both scales without the other’s help. They picked the C-eHEALS only due to its fewer item numbers. They had lower education and household income, town or village residence, short Internet use history, and had never used the Internet for health information. Their eHealth literacy scores on both scales were low.

Of the eight interviewees who preferred to use the C-DHLI in practical settings, most stated the C-DHLI was easier to understand as it had more **specific descriptions**. Specifically, they explained that they felt the C-DHLI items described more distinct skill categories, while items of C-eHEALS require long time for them to comprehend. For example, the first three items of C-eHEALS were about “what health resources…”, “where to find…”, and “how to find…”. However, when they were translated into Chinese, some interviewees’ first impression was that their meanings were similar, so it took more time for them to understand their differences. The same happened for item 4 (“…how to use the Internet to answer my health questions”) and item 5 (“…how to use health information to help me”) of the C-eHEALS.


*“I have difficulty distinguishing the items of C-eHEALS and feel some of them quite similar. Thus, when answering the questions, I just roughly choose an option. But for the C-DHLI, I can tell the difference between items with specific scenarios, so I can exactly understand what this scale would like to ask.”*

*[Interviewee B10]*


In addition, some—particularly those with lower education (e.g., middle school) and shorter Internet use history—described **less semantic complexity** and **shorter length** of the C-DHLI’s items as reducing their burden in answering the scale. Additionally, *Interviewee B7* also emphasized the higher readability of C-DHLI than C-eHEALS, not only about the items but also about item options,


*“C-DHLI’s items are shorter, easier, and more straightforward. Moreover, the answer options of C-DHLI regarding difficulty or frequency are more objective and easier to answer than eHEALS’s options about the agreement.”*

*[Interviewee B7]*


Another two interviewees (i.e., *C14* and *C15*) did not give an explicit preference for either scale. They explained that they rarely used the Internet for health information and eHealth literacy was unimportant in their daily life. They commented the C-DHLI is relatively easier to understand but more time-consuming.

#### 3.4.2. Content Comprehensiveness

At the beginning of the interview, the definition of eHealth literacy was introduced to each interviewee. After that, they were asked to comment on the extent which the two scales have captured the concept. Interviewees mentioned that the C-DHLI provided a more comprehensive description about health-related Internet skills/contexts than the C-eHEALS.


*“I think the C-DHLI **covers all skills/scenarios** that I would potentially encounter on the Internet, and some I have not even encountered yet…”*

*[Interviewee B8]*


Some interviewees indicated that the **clear structure** of C-DHLI increased the clarity of what skills eHealth literacy aimed to measure, whereas the C-eHEALS described more about an overall perception of eHealth use.


*“The subtitles of C-DHLI help improve its structure, so I can know what specific skills it would like to measure. The C-DHLI also help me understand which skills I need to improve in the future.”*

*[Interviewee B6]*


However, *Interviewee A3* also complained about the length of C-DHLI:


*“I think eHEALS already reflects the definition of eHealth literacy, whereas the C-DHLI is a bit too long and may contain some useless information.”*

*[Interviewee A3]*


## 4. Discussion

This study showed that both C-eHEALS and C-DHLI were reliable tools in assessing the eHealth literacy of Chinese older adults. The C-DHLI showed itself to be more valid than the C-eHEALS with better construct validity based on the nine a-priori hypotheses. Additionally, qualitative results revealed that the C-DHLI was preferred by interviewees across different backgrounds, since it was perceived as more readable (e.g., having a clear structure, specific descriptions, shorter sentence length, and less semantic complexity) than C-eHEALS. The C-eHEALS was more favored by those who can easily understand it for having fewer items (and therefore being less time-consuming) than the C-DHLI.

The quantitative results showed that both C-eHEALS and C-DHLI had good internal consistency and test-retest reliability in Chinese older adults. The result was consistent with the findings of a Korean study which validated the Korean version of DHLI and eHEALS in older adults (Cronbach’s alpha K-DHLI: α = 0.93, K-eHEALS: α = 0.90; ICC K-DHLI: 0.844, K-eHEALS: 0.769) [13]. As found in previous studies, eHEALS demonstrated good readability in older adults in Iran, the US, and Norway [38,39,40,41], with Cronbach’s α ranging from 0.85 to 0.99. As the DHLI was developed in 2017 and has only been validated in older adults in South Korea, this study expands evidence that the DHLI is an internally consistent and temporally stable measure for the elderly’s eHealth literacy assessment in the Chinese context. Researchers and healthcare professionals can consider both C-eHEALS and C-DHLI to assess Chinese older adults’ ability to utilize online health information for health purposes in practice settings.

Overall, the C-DHLI demonstrated better construct validity than the C-eHEALS in the older Chinese population. Specifically, all nine hypotheses were fulfilled in the C-DHLI, and five of them were fulfilled in the C-eHEALS. In addition, for the five hypotheses fulfilled in both scales, the correlations between C-DHLI and the hypothesized factors (i.e., education, occupation, self-rated Internet skills, using the Internet for health information, and health literacy) were stronger than such with the C-eHEALS scores. As stated before, the disparities in social structure, digital divide, and health literacy are related to older adults’ eHealth literacy, which would further exacerbate health disparities [8]. Based on the results, the C-DHLI can better reflect the impact of such disparities on eHealth literacy among Chinese older adults compared with the C-eHEALS. Some possible explanations for the findings were proposed: First, the C-eHEALS measures one’s overall perception of eHealth use by eight items, while the C-DHLI contains seven specific skill categories on eHealth use, with twenty-one items in total. It was found that multi-item measures were likely to be more reliable and valid (e.g., [42]). As eHealth literacy is a multi-dimensional construct, the overall measure (i.e., C-eHEALS) may be less significant in construct validity than the multi-dimensional measure (i.e., C-DHLI). Second, the C-DHLI is developed based on one’s actual performance on eHealth usage. Hence, its items are specific skill orientations (e.g., operational skills and information evaluation) closely related to older adults’ basic knowledge and skills. Thus, the C-DHLI may demonstrate stronger correlations with the hypothesized construct than the C-eHEALS, which emphasizes one’s self-efficacy in eHealth usage. This point was further demonstrated in the present study from the ninth construct hypothesis regarding health literacy. Functional literacy, as one dimension of health literacy, reflects the basic reading and understanding skills of health information [43]. This study found functional literacy was moderately correlated with the C-DHLI score (*r* = 0.402, *p* < 0.001) while not significantly associated with the C-eHEALS score. Some previous studies have criticized the weak relationship between one’s self-efficacy and actual performance on eHealth use [16,17]. Although eHEALS has been validated in older populations, its validity may warrant more discussions.

The results from the qualitative data confirm that a clear difference of eHealth literacy levels exists within older individuals, potentially influenced by their socio-demographic characteristics and Internet use history/skills [21]. Respondents’ perceptions of the scale are also influenced by their background, especially their education levels, affecting their preference in choosing eHealth literacy assessment in practice. Based on the qualitative results, the C-eHEALS may be preferred by those (especially those with higher educational attainments and SES) who can easily understand both scales. They tend to choose the one with fewer items (i.e., C-eHEALS) since it takes less time to complete. However, most interviewees (across different background levels) preferred to use the C-DHLI, as they perceived it easier to understand and more comprehensive than the C-eHEALS. As many older adults are beginners to health technology, the clear structure and specific skill/scenario description of the C-DHLI can help them navigate through the scale. Moreover, they also found that shorter sentence length and lower semantic complexity of its items decreased their response burden. The results suggest that respondents would prefer to use a scale which is perceived to be more readable. The qualitative findings may indicate that the C-eHEALS can be applied to certain groups of elderly, such as those with higher education and SES, but the C-DHLI is more acceptable to the general older population.

This study confirmed the reliability and validity of the two eHealth literacy instruments in Chinese older adults. However, as mentioned before, as the digital health technologies have advanced a lot in recent years (e.g., the popularity of social media), refinement of the instrument is therefore warranted, for example, to capture the characteristics of online interactions. In addition, the C-DHLI, as a newly developed instrument, has been translated into Chinese in 2021 by our research team and validated in 277 Chinese older adults for the first time. More evidence regarding its psychometric properties in larger and more representative samples in China is also needed. There are several limitations of the study that should also be noted. First, the present study was a web-based survey with the sample being Chinese older adults who voluntarily participated in this study. Selection bias may exist as the participants were older adults who had experience with the Internet and were more likely to be interested in accessing health information online. Second, reporting bias (e.g., social desirability and recall bias) may exist as the questionnaire was self-reported and self-administered. Third, nine a-priori hypotheses were used to examine the construct validity in the present study, which is often considered less powerful than criterion validation. However, with strong theories and specific expectations, it is possible to acquire substantial evidence that the measurement instrument is measuring what it purports to measure [44].

## 5. Conclusions

To conclude, the Chinese versions of eHEALS and DHLI were reliable in assessing eHealth literacy, but the C-DHLI was more valid than the C-eHEALS among Chinese older adults. In addition, qualitative results showed that the C-DHLI was more favored by respondents with different backgrounds, due to its clear structure, specific descriptions, shorter sentence length, and lesser semantic complexity. The C-eHEALS was preferred by people who can easily understand it due to its brevity (i.e., fewer item numbers). The findings suggest that the C-DHLI may be more applicable to general older populations in China for eHealth literacy assessment; the C-eHEALS can be considered in certain elderly groups (e.g., those with higher education or SES) in daily practice. In addition, the refinement of the eHEALS is still strongly recommended to capture the evolution of digital health technologies over time.

## Figures and Tables

**Table 1 ijerph-20-03293-t001:** Characteristics of the participants (N = 277).

Characteristics	N (%)	eHEALS Score		DHLI Score	
		Mean ± SD	*p* Value	Mean ± SD	*p* Value
* **Socio-demographic variables** *					
Age, years			0.740		0.002
55–59	167 (60.3)	3.5 ± 0.59 (ref)		2.59 ± 0.48 (ref)	
60–69	91 (32.9)	3.52 ± 0.65		2.49 ± 0.59	
70–79	19 (6.9)	3.4 ± 0.78		2.15 ± 0.63 **	
Gender			0.442		0.118
Male	142 (51.3)	3.53 ± 0.62		2.58 ± 0.53	
Female	135 (48.7)	3.47 ± 0.63		2.48 ± 0.55	
Type of residence			0.701		<0.001
Town/village	136 (49.1)	3.51 ± 0.56		2.40 ± 0.53	
City	141 (50.9)	3.48 ± 0.68		2.65 ± 0.52	
Education			0.006		<0.001
Primary school or below	41 (14.8)	3.23 ± 0.70 (ref)		1.99 ± 0.49 (ref)	
Middle or High school	163 (58.8)	3.52 ± 0.62 **		2.52 ± 0.52 ***	
Junior college or above	73 (26.4)	3.61 ± 0.56 **		2.84 ± 0.34 ***	
Marital status			0.070		0.114
Married	256 (92.4)	3.52 ± 0.61		2.54 ± 0.54	
Unmarried/widowed/others	21 (7.6)	3.26 ± 0.71		2.35 ± 0.48	
Occupation (now or before retirement)			0.025		<0.001
Unemployed	44 (15.9)	3.27 ± 0.64 (ref)		2.33 ± 0.62 (ref)	
Farmer/trader/others	150 (44.8)	3.52 ± 0.62		2.43 ± 0.52 ***	
Government workers/professionals	109 (39.4)	3.57 ± 0.60 **		2.72 ± 0.46 ***	
Monthly household income, RMB			0.405		<0.001
Under 2500	51 (18.4)	3.42 ± 0.66 (ref)		2.26 ± 0.58 (ref)	
2500–5000	94 (33.9)	3.48 ± 0.61		2.35 ± 0.51	
5000–10,000	91 (32.9)	3.59 ± 0.52		2.72 ± 0.39 ***	
Over 10,000	41 (14.8)	3.46 ± 0.82		2.85 ± 0.54 ***	
* **Internet use-related variables** *					
Internet use history			0.144		<0.001
5 years or below	96 (34.7)	3.41 ± 0.64 (ref)		2.26 ± 0.60 (ref)	
5 to 10 years	82 (29.6)	3.50 ± 0.68		2.57 ± 0.48 ***	
Over 10 years	99 (35.7)	3.58 ± 0.55		2.76 ± 0.39 ***	
Self-rated Internet skills			<0.001		<0.001
Poor or fair	83 (30)	3.25 ± 0.67 (ref)		2.10 ± 0.53 (ref)	
Average	167 (60.3)	3.6 ± 0.56 ***		2.69 ± 0.37 ***	
Good or excellent	27 (9.7)	3.61 ± 0.65 **		2.87 ± 0.66 ***	
Using Internet for health information			<0.001		<0.001
Yes	220 (79.4)	3.60 ± 0.59		2.66 ± 0.45	
No	57 (20.6)	3.12 ± 0.62		2.03 ± 0.56	

Note: ** *p* < 0.01, *** *p* < 0.001 for multiple group comparison using Fisher’s LSD method (One-way ANOVA).

**Table 2 ijerph-20-03293-t002:** Correlations between two eHealth literacy scales and construct variables (N = 277).

Variables	eHEALS Score	DHLI Score
* **Socio-demographic factors** *		
Age	−0.020	−0.217 ***
Type of residence (1 = town/village, 2 = city)	0.028	0.240 ***
Education (1 = primary school or below, 2 = middle/high school, 3 = junior college or above)	0.179 **	0.479 ***
Occupation (1 = unemployed, 2 = trader/labor/farmer/others, 3 = government workers/professionals)	0.150 *	0.298 ***
Household income (1 = under 2500, 4 = over 10,000)	0.088	0.403 ***
* **Internet use-related factors** *		
Internet use history (1 = 5 years or below, 3 = over 10 years)	0.115	0.346 ***
Self-rated Internet skills (1 = poor/fair, 3 = good/excellent)	0.242 ***	0.492 ***
Using the Internet for health information (0 = no, 1 = yes)	0.328 ***	0.432 ***
Health literacy (HLS-14)	0.391 ***	0.562 ***
Factor 1: Functional literacy	0.100	0.402 ***
Factor 2: Interactive literacy	0.407 ***	0.389 ***
Factor 3: Critical literacy	0.381 ***	0.349 ***

* *p* < 0.05, ** *p* < 0.01, *** *p* < 0.001.

**Table 3 ijerph-20-03293-t003:** Characteristics and qualitative results of interviewees (N = 15).

Interviewee	Socio-Demographic Characteristics						Internet Use		Scale Score and Preference				
	Age	Gender	Education	Occupation (now or before Retirement)	Household Income (RMB/Month)	Residence Type	Internet Use History (Year)	Using Internet for Health Information	C-eHEALS Mean Score (1–5)	C-DHLI Mean Score (1–4)	Perceived Readability of C-eHEALS	Perceived Readability of C-DHLI	Preference to Use which Scale
A1	58	Male	High school	Government worker	5000–10,000	City	12	yes	3.75	3.11	Easy	Easy	eHEALS
A2	56	Male	Junior college	Government worker	5000–10,000	City	18	yes	3.875	2.28	Easy	Easy	eHEALS
A3	63	Male	Junior college	Trader	5000–10,000	City	21	yes	4	3.17	Easy	Easy	eHEALS
A4	55	Female	Primary school	Unemployed	Lower than 2500	Town	3	No	2.125	2	Very difficult	Very difficult	eHEALS
A5	71	Female	Primary school	Trader	Lower than 2500	Village	1	No	1	1.06	Very difficult	Very difficult	eHEALS
B6	56	Female	High school	Government worker	2500–5000	Town	10	yes	3.625	2.89	Rather difficult	Easy	DHLI
B7	55	Female	Junior college	Government worker	5000–10,000	City	15	yes	3.75	2.61	Rather Easy	Easy	DHLI
B8	60	Male	Junior college	Government worker	5000–10,000	City	18	yes	4	2.94	Rather Easy	Easy	DHLI
B9	68	Female	Junior college	Professional	5000–10,000	City	10	yes	3.625	2.72	Rather difficult	Easy	DHLI
B10	59	Male	Middle school	Trader	5000–10,000	Town	10	yes	3.875	2.72	Rather difficult	Rather Easy	DHLI
B11	61	Male	Middle school	Trader	2500–5000	Town	6	yes	3.25	2.5	Rather difficult	Rather Easy	DHLI
B12	56	Female	Middle school	Trader	2500–5000	Town	8	yes	3.5	2.44	Difficult	Rather Easy	DHLI
B13	61	Female	High school	Government worker	2500–5000	Town	8	yes	3.25	2.17	Difficult	Rather difficult	DHLI
C14	57	Female	Middle school	Farmer/labor	2500–5000	Town	7	yes	3.25	2.17	Difficult	Rather difficult	NS
C15	65	Female	Primary school	Unemployed	2500–5000	Town	6	No	2.625	2	Very difficult	Difficult	NS

Note: eHEALS = eHealth literacy scale, DHLI = digital health literacy instrument, NS = not sure.

## Data Availability

The data presented in this study are available on request from the corresponding author. The data are not publicly available due to respondents’ privacy.

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
