# Peer review of "Comparison of eHealth Literacy Scale (eHEALS) and Digital Health Literacy Instrument (DHLI) in Assessing Electronic Health Literacy in Chinese Older Adults: A Mixed-Methods Approach"

_ijerph, 2023, doi:10.3390/ijerph20043293_

Round 1
Reviewer 1 Report
1. The second paragraph of the Introduction took the form of “If not... there is a risk that...” to describe the importance of eHealth literacy for older adults. However, eHealth literacy is not a direct determinant of physical health problems. Therefore, maybe the authors should consider using a factual statement such as “People who are eHealth literate will ......”.
2. Section 2.1 of the manuscript should describe in detail the sampling process of the 277 participants and elaborate on the representation of these participants at the social level.
3. In the Data collection section, the authors briefly summarized the questionnaires used in the study, which would have been better described in detail using the heading “Measurement Tools” or “Measurement”.
4. Although the authors stressed the possible bias of sample selection as one of the limitations, the representative of the sample is the most important issue in such an instrument validation study. Therefore, it is still recommended that the authors should elaborate on the sampling design in the Method.
5. In the Discussion and Conclusion section, the authors can further point out the direction of revision of these scales or the design ideas of scales suitable for China based on the results of the qualitative analysis.
Reviewer 2 Report
Summary: In this paper, authors compared two well-established internet health literacy instruments for older adults i.e., Chinese versions of eHealth literacy scale (C-eHEALS) and digital health literacy instrument (C-DHLI). The authors claimed that most interviewees perceived the C-DHLI as more readable than C-eHEALS for its clear structure, specific description, short sentence length, and less semantic complexity.
The paper is written in a very good and organized manner, but there are some (following) clarification/ suggestions are needed to be addressed accordingly.
1) In line 141, “The recruitment stopped when theoretical saturation emerged” what does mean theoretical saturation?
2) In line 151, “Cronbach's α with a value > 0.7 indicating acceptable level”. Author needs to clarify or referenced this statement that why and how an acceptable level is based on the value of Cronbach's α should be greater than 0.7?
3) How interviewees are labeled as A1, B6 and C14. Why different alphabets? Did authors categorize them? If yes then on what basis.
4) Add a comparison table with other studies such as Korean, Iranian, US and Norwegian studies.
5) Authors are suggested please proofread the paper again because there are some minor spelling / typos / issues such as
a. In line 213, confirmed the a-prior-defined hypotheses (the should be removed)
b. In line 312, The result was consistent with the findings of a Korean study which validated the Korean version of DHLI (add reference of Korean study)
c. In line 315. As found in previous studies, eHEALS demonstrated good readability in older adults in Iran, the US, and Norway (add references of these studies)
d. In line 334, eHealth use using eight items, word using should be deleted.
6) Also update the related work on older adults with latest and relevant work such as follows
Green G. Seniors' eHealth literacy, health and education status and personal health knowledge. Digit Health. 2022 Mar 27;8:20552076221089803. doi: 10.1177/20552076221089803
A. Mehmood, A. Nadeem, M. Ashraf, M. S. Siddiqui, K. Rizwan and K. Ahsan, "A Fall Risk Assessment Mechanism for Elderly People Through Muscle Fatigue Analysis on Data From Body Area Sensor Network," in IEEE Sensors Journal, vol. 21, no. 5, pp. 6679-6690, 1 March1, 2021
Decision: Accepted with minor changes
